# Modification of Fire Regimes Inferred from the Age Structure of Two Conifer Species in a Tropical Montane Forest, Mexico

**Jesús Eduardo Sáenz-Ceja [1,2] and Diego Rafel Pérez-Salicrup [2,*]** 

[1]   Graduate Program in Biological Sciences, Universidad Nacional Autónoma de México,
      Circuito de Posgrados, Coyoacán, Mexico City 04510, Mexico; jsaenz@cieco.unam.mx
[2]   Instituto de Investigaciones en Ecosistemas y Sustentabilidad, Universidad Nacional Autónoma de México,
      Antigua Carretera a Pátzcuaro 8701, Ex Hacienda San José de la Huerta, Morelia 58190, Mexico
*    Correspondence: diego@cieco.unam.mx

**Abstract:** Research Highlights: Age structure was used to infer fire regimes in the Monarch Butterfly Biosphere Reserve. Uneven-aged structures in stands dominated and co-dominated by pine and fir species, which are distributed according to an altitudinal gradient, indicated a regime of frequent, low-severity, and low-intensity fires. Background and Objectives: Age structure analyses have been used to infer natural and disrupted fire regimes when field-based descriptions of fires are scarce or unavailable. In montane conifer forests, fire regimes typically vary according to an altitudinal gradient, shaping contrasting tree establishment patterns. In the Monarch Butterfly Biosphere Reserve, Mexico, the altitudinal distribution and fire regimes of sacred fir forests (*Abies religiosa*), smooth-bark Mexican pine forests (*Pinus pseudostrobus*), and mixed-conifer forests are poorly documented. The objectives of this study were to determine the altitudinal ranges occupied by mono-dominant and co-dominant stands and to reconstruct tree establishment history to infer historical fire regimes. Materials and Methods: Six altitudinal transects were established along the reserve, each one at elevations from 2400 to 3300 m, with sampling sites at every 150 m of elevation. In each site, increment cores were collected from the base of 25 mature trees. A total of 800 increment cores were collected and cross-dated. Results: *P. pseudostrobus* is dominant in stands between 2400 and 2850 m, *A. religiosa* between 3150 and 3300 m, and both species co-dominate between 2850 and 3150 m. The establishment pattern for both species has been continuous, represented by uneven-aged structures, suggesting that tree establishment in smooth-bark Mexican pine forests, mixed-conifer forests, and sacred fir forests, is likely to be associated with frequent, low-severity, and low-intensity fires. Conclusions: These fire regimes suggest, by the one hand, the disruption of natural fire regimes by human activities, limiting the occurrence of high-severity fires; on the other hand, a distinctive feature of these tropical montane forests.

**Keywords:** conifer forest; dendrochronology; disturbance regime; Monarch Butterfly Biosphere Reserve; stand regeneration

---

## 1. Introduction

Age structure is the distribution of ages for all individuals in a population [1], which reflects both the establishment of new individuals in a population over time [2] and the ability of individuals to survive under past environmental conditions [3]. The age structure can be associated with disturbance regimes since an age distribution is the result of tree mortality and the initiation of new cohorts following disturbances [4]. In this context, an uneven-aged distribution, representing a continuous

regeneration pattern, may be related to frequent disturbances, whereas an even-aged distribution, representing pulses of regeneration, could be associated with episodic and large disturbances [5].

Fire is considered a key disturbance that influences tree regeneration, establishment patterns, and age structure in conifer forests [6,7]. Based on an altitudinal gradient of temperature and precipitation, many montane conifer forests typically experience a fire regime continuum, with contrasting fire regimes at the extreme elevations [8]. For instance, at low elevations, many forests dominated by *Pinus* sp. experience frequent, low-severity, and low-intensity fires, which in turn are reflected in continuous tree establishment [1]. In contrast, at high altitudes, forests dominated by subalpine species, such as *Abies* sp., generally exhibit infrequent, high-intensity, and high-severity fires [8], which lead to a stand replacement, with an age structure that suggests synchronous or pulsed establishment after each fire [9].

Conifer forests distributed along the Trans-Mexican Volcanic System in Mexico are considered relicts of the last glacial period [10] and are very diverse in conifer species [11]. At high-altitudes, where humid and cold conditions are found, fir species (*Abies* spp.) dominate, whereas at lower altitudes with drier and warmer conditions, pines are dominant (*Pinus* spp.) [12]. Information on fire dynamics and tree demography in these tropical montane conifer forests are still very scant, in part because forest legacies and stumps decay faster than in higher-latitude forests, limiting the elaboration of long fire-scar chronologies [13].

The Monarch Butterfly Biosphere Reserve (MBBR) is located in central Mexico, where millions of monarch butterflies (*Danaus plexippus* Linnaeus) hibernate annually from November to March [14], who generally arrive most often to the same sanctuaries, located mainly in *Abies religiosa* (H.B.K.) Schl. et Cham. (sacred fir) and mixed-conifer stands, at elevations above 3000 m [15]. The butterflies form compact colonies within mature trees, perching on the branches, whose canopy moderate extreme temperatures and allows butterflies to be protected from freezing [16]. This reserve includes *Pinus pseudostrobus* (smooth-bark Mexican pine) forests at low elevations [17]. However, during the last 20 years, illegal logging, livestock grazing, and fires have degraded the conifer forests of the reserve [18].

Fire regimes in the MBBR may differ according to an altitudinal gradient, as documented in montane tropical forests such as those located in the US Southwest [8,19]. On the one hand, sacred fir forests may experience stand-replacement fires, since fuel loads are greater than in montane subtropical pine forests [20], and sacred fir trees form dense canopies and have low crown base height [21]. On the other hand, smooth-bark Mexican pine forests are likely to experience surface fires, whose trees have thick bark, a typical trait of forest adapted to frequent low-severity fires [22]. However, there are pieces of evidence that natural fire regimes may have been altered by human disturbances, which in turn can modify tree regeneration patterns [23].

Fire regime alteration could considerably modify forest structure, fuel load dynamics, and tree establishment patterns [24,25]. An increase in fire return intervals could result in higher tree densities and accumulation of fuel loads, which in turn can trigger high-severity fires during drought periods [26]. Alternatively, a reduction of fire intervals, either by drier conditions associated with global warming or by an increase of human ignition sources could decrease tree recruitment and change stand dominance [27,28]. Hence, it is fundamental to understand the role of fire and climate on the regeneration of conifer forests in the MBBR.

An important limitation for the assessment of fire regimes in the conifer forests of the MBBR is the lack of field-based records of past disturbances, such as fires, windstorms, insect breakouts, and logging [29], which makes it difficult to establish a clear relationship between age structure and frequency of disturbances. However, since the age structure of montane conifers is strongly correlated with fire return intervals, as found in ponderosa pine forests in the US Southwest and northern Mexico [30,31], the age structure of sacred fir and smooth-bark Mexican pine could likely give information about fire return intervals in the MBBR [32,33].

The altitudinal intervals occupied by sacred fir forests, mixed-conifer forests, and smooth-bark Mexican pine forests are currently unknown, but they may represent distinct microclimatic and soil

conditions, where ecosystem processes, species dominance, floristic elements, and stand structures, might be associated with different fire regimes [21,29]. In addition, a quantification of the altitudinal distribution and the age structure of these conifers would allow identifying a migration of these species to higher elevations in response to drier conditions associated with climate warming, in which *P. pseudostrobus* could replace *A. religiosa* [34]. Moreover, at intermediate elevations in mixed-conifer forests, the analysis of age structure could provide information on conifer succession [35], in particular, whether *P. pseudostrobus*, a shade-intolerant species, establish previously to *A. religiosa*, a shade-tolerant species, as proposed for these montane tropical conifer forests [36].

The objectives of this study were (1) to determine the altitudinal intervals occupied by dominant and co-dominant stands of *A. religiosa* and *P. pseudostrobus* in the MBBR, (2) to characterize the establishment patterns of *A. religiosa* and *P. pseudostrobus* and determine whether they differ between mono-dominant and co-dominant stands, and (3) to evaluate whether the age structure in conifer stands is consistent with fire regimes proposed for the dominant conifer species. This is one of the first contributions to understand disturbance dynamics in Mexican tropical montane conifer forests and will provide a first approximation of the fire regimes experienced in this charismatic field site. This information will be useful in the management and long-term preservation of the ecological integrity of the habitat of the monarch butterflies.

## 2. Materials and Methods

### 2.1. Study Site

The MBBR has a total surface of 56,256 ha, located between the Mexican States of Michoacán and México. It is part of the Trans-Mexican Volcanic Belt and formed mainly of volcanic cones and hills, with elevations ranging from 2200 to 3640 m [37]. The reserve includes three main mountain massifs in whose peaks the overwintering colonies of monarch butterflies are located [18]: Sierra de Chincua in the northern zone, Sierra Campanario-Chivati-Huacal in the central zone, and Sierra Cerro Pelón in the southern zone (Figure 1). Climate is temperate subhumid with a rainy season from July to October, mean annual temperatures from 8 to 22 °C, and mean total annual precipitation of 700 mm in the southern zone and 1250 mm in the northern and central zones [38].

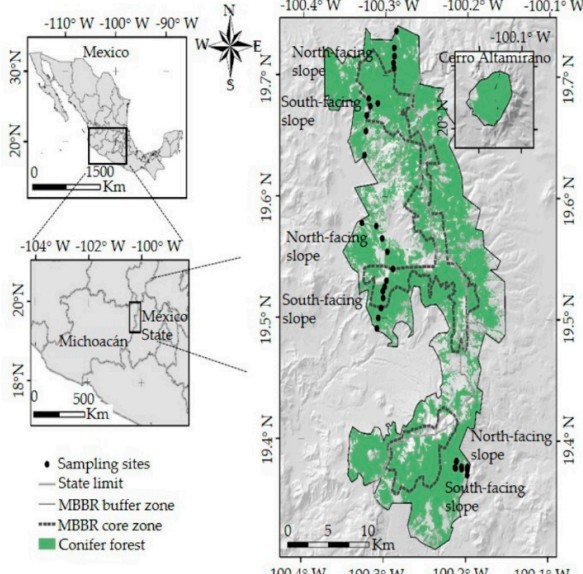

**Figure 1.** Location of the Monarch Butterfly Biosphere Reserve, central México, and the 32 sampling sites along six altitudinal transects.

The MBBR is covered by four major vegetation types: conifer forest, oak forest, cloud forest, and montane grasslands. Conifer forests consist of *A. religiosa* mono-dominated forests (Figure 2a), *A. religiosa–P. pseudostrobus* co-dominated forests (Figure 2b), and *P. pseudostrobus* mono-dominated forests (Figure 2c) [39]. The habitat of *A. religiosa* is restricted to high-elevation mountains and deep ravines, with frequent fog occurrence, low insolation, high moisture, and soils with high organic matter [40]. Meanwhile, *P. pseudostrobus* is found in lower-elevation slopes, with drier conditions and deeper soils [41].

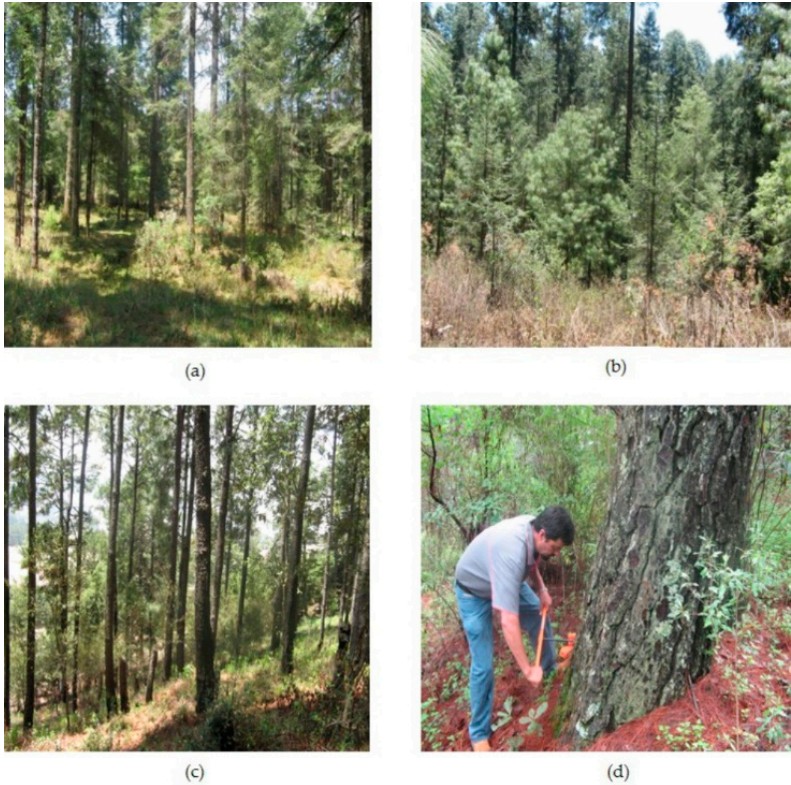

**Figure 2.** Stands dominated by *A. religiosa* (**a**), co-dominated by *A. religiosa* and *P. pseudostrobus* (**b**), dominated by *P. pseudostrobus* (**c**), and extraction of increment cores (**d**).

The extent of the MBBR has changed over time. In 1986, the surrounding area of the historical colonies of monarch butterflies was protected by a first decree (16,110 ha), which covered mainly sacred fir stands. In 2000, a second decree included the surrounding pine and mixed conifer forests located at lower altitudes and increased its extent to 56,256 ha. Currently, 24% of the MBBR area includes three core zones (13,552 ha), where commercial logging is forbidden, but scientific research and salvage logging after fires, bark beetle detections, and windstorms are allowed. The rest of the area includes two buffer zones, where commercial logging, mining, livestock grazing, and agricultural activities are allowed under environmental restrictions [18]. The MBBR is inhabited by 27,000 people, settled mostly in rural communities [42]. Despite the protected status of the MBBR, most of the land is owned by these rural communities in the form of communal lands (*ejidos*), a particular social land tenure, legacy of the Mexican Revolution, and indigenous communities. Forests in both of these land tenure systems are managed by common agreements in assemblies [43]. Currently, the territory of the MBRR belongs to 57 *ejidos*, 13 indigenous communities, one federal property, and one state property, as well as several small private properties [44].

There is a paucity of historical information on human and natural disturbances, which includes windthrows, fires, bark beetle occurrence, selective logging, firewood collection, livestock grazing, and resin extraction resin extraction of pine trees [45,46]. Pine forests at lower elevations in the southern

zone have recently transformed into avocado plantations. Sustainable forest management, however, has been implemented in several locations, and the rate of deforestation has decreased in the last decade [18]. Large disturbances like insect outbreaks and fires >40 ha have not been documented in the conifer forest in the MBBR in the last 30 years [47,48]. Only two studies reported high-magnitude fires: a large fire dated in 1670 through paleoecological techniques in a mixed-conifer forest of Sierra Chincua [49], and a 300-ha fire located in sacred fir forests of Sierra Cerro Pelón [50]. Most fires occur from March to June, during the dry season, and are associated with human sources of ignition, such as agricultural activities, forestry, campfires, and conflicts between communities. Fire is used by rural inhabitants in agricultural activities, to promote grasslands, non-timber forest products, and for clearing roadsides [51].

### 2.2. Sampling Design

The reserve area was divided into three zones: northern, central, and southern. In each zone, remote sensing imagery and geographical information systems were used to locate areas with continuous (not fragmented) conifer forest cover along an altitudinal gradient between 2400 and 3300 m, which corresponds to the extreme altitudinal ranges reported for *P. pseudostrobus* and *A. religiosa* [34,52]. Six altitudinal transects were established, two in each zone of the reserve. Within each zone, one transect was established in a North-, and the other one in a South-facing slope, as slope orientation might influence the establishment of both species. In South-facing slopes of this region, trees receive more insolation than North-facing slopes, which could promote the establishment of *P. pseudostrobus*, a more drought-sensitive species than *A. religiosa* [53].

In each transect, sampling points at every 150 m of elevation were established (Figure 1). Not all transects covered the entire altitudinal gradient (2400 to 3300 m) (Table 1) due to the following reasons: (1) in the northern and southern zones, the mountains do not reach the highest evaluated elevation (3300 m), (2) in the northern zone and North-facing slopes, forests at 2400 m of elevation are dominated by *Quercus* sp. and are outside of the MBBR boundaries, and (3) in the southern zone, the establishment of sampling sites was only possible at elevations higher than 2700 m because, at lower elevations, conifer forests were previously deforested and in most cases converted to avocado plantations. The final number of sampling sites was 32.

**Table 1.** Location of sampling sites in the Monarch Butterfly Biosphere Reserve (MBBR), according to zone, facing-slope, and altitude.

| Zone | Facing-Slope | Altitude (m) | Latitude (North) | Longitude (West) | Facing Slope | Altitude (m) | Latitude (North) | Longitude (West) |
|------|------|------|------|------|------|------|------|------|
| Northern | North | 3150 | 19.706 | −100.289 | South | 3150 | 19.677 | −100.309 |
| | | 3000 | 19.710 | −100.290 | | 3000 | 19.681 | −100.320 |
| | | 2850 | 19.716 | −100.289 | | 2850 | 19.674 | −100.318 |
| | | 2700 | 19.723 | −100.289 | | 2700 | 19.667 | −100.322 |
| | | 2550 | 19.736 | −100.287 | | 2550 | 19.654 | −100.323 |
| | | 2400 | - | - | | 2400 | 19.634 | −100.325 |
| Central | North | 3150 | - | - | South | 3150 | 19.530 | −100.297 |
| | | 3000 | 19.540 | −100.289 | | 3000 | 19.525 | −100.299 |
| | | 2850 | 19.554 | −100.296 | | 2850 | 19.522 | −100.301 |
| | | 2700 | 19.565 | −100.302 | | 2700 | 19.516 | −100.301 |
| | | 2550 | 19.576 | −100.310 | | 2550 | 19.508 | −100.304 |
| | | 2400 | 19.578 | −100.327 | | 2400 | 19.491 | −100.307 |
| Southern | North | 3300 | 19.377 | −100.213 | South | 3300 | 19.376 | −100.212 |
| | | 3150 | 19.382 | −100.211 | | 3150 | 19.382 | −100.211 |
| | | 3000 | 19.378 | −100.205 | | 3000 | 19.376 | −100.204 |
| | | 2850 | 19.378 | −100.197 | | 2850 | 19.377 | −100.198 |
| | | 2700 | 19.375 | −100.198 | | 2700 | 19.371 | −100.198 |

In each sampling site, the nearest 25 *P. pseudostrobus* and *A. religiosa* trees with a diameter at breast height (DBH = 1.3 m) ≥ 25 cm were selected for coring. We identified each tree to species, measured DBH in all trees, and then sampled 1 to 3 increment cores, depending on whether the pith was reached in the first, second, or third attempt. Increment cores were extracted from the downslope side of the stem base (Figure 2d) because it allowed us to extract increment cores with wider and more visible

tree-rings due to the formation of reaction wood in the downslope side of conifer trees [54]. Trees were cored as close to the ground as possible to minimize uncertainty in age estimates associated with the time it takes to reach coring standard height. A total of 800 increment cores were collected.

Samples for all trees were sanded and cross-dated, and the year of establishment for each tree was determined following standard dendrochronological techniques [55,56]. Skeleton-plots were made for each sample and compared with a master chronology for the MBBR [57] to identify false rings, missing rings, and micro-rings. Tree-rings were counted under a microscope.

### 2.3. Data Analyses

Sampling points where ≥60% of trees belonged to one species were considered to be dominated by that species. Points, where *P. pseudostrobus* and *A. religiosa* included 40–59% of individuals, were considered co-dominant. A generalized linear model (GLM) was used to evaluate the effect of the zone (northern, central, and southern), slope orientation (north and south), and elevation on the proportion of trees of these two species. Once the altitudinal distribution intervals of *A. religiosa* and *P. pseudostrobus* were obtained, the potential surface of the MBBR where both species are dominant or co-dominant was estimated.

Tree age was analyzed using a GLM (glm function in the stats package) to evaluate differences in age according to elevation, zone, and slope aspect. To identify differences in age between both species, a Wilcoxon-test (wilcox.test in the stats package) was conducted. Analyses were conducted in R version 3.4.3 [58]. Due to the short length of our age data, they were composited according to elevation and plotted into 5-year classes (periods) according to their pith dates.

Establishment by pulses was defined through the timeline when, (1) there were one or two 5-year periods without tree recruitment between the first and the last 5-year period that recorded tree recruitment, (2) a period when the recorded establishment was more than 10 trees. The continuous establishment was defined when between the first and the last 5-year periods that recorded tree recruitment, there were no 5-year periods without new trees within the population.

The tree establishment years were compared graphically with the Palmer Drought Severity Index (PDSI) obtained for central Mexico [59], to evaluate whether establishment pulses were associated with wet years [30].

## 3. Results

### 3.1. Altitudinal Ranges of Conifer Dominant Species

A total of 32 points were sampled, 18 of which were dominated by *P. pseudostrobus*, four by *A. religiosa*, and 10 co-dominated by both species. Of all collected cores, 71% were *P. pseudostrobus* and 29% *A. religiosa*. The proportion of these two species was affected by the altitude for both, *A. religiosa* ($R^2 = 0.63$, df = 31, F = 59.69, $p < 0.05$) and for *P. pseudostrobus* ($R^2 = 0.61$, df = 31, F = 52.87, $p < 0.05$), according to the GLM-test. Stands dominated by *A. religiosa* were found at elevations above 3150 m, co-dominated stands between 2850 and 3150 m, and stands dominated by *P. pseudostrobus* below 2850 m (Figure 3). A proportional higher abundance of *P. pseudostrobus* in the northern zone (84% of the samples), and a higher proportion of stands dominated by *A. religiosa* in the southern zone (53% of the samples) were detected. The central zone had the lowest altitudinal distribution of *A. religiosa* (2700 m) and had a greater abundance of *P. pseudostrobus* at 3150 m (76% of the total trees).

According to the altitudinal ranges associated with mono- and co-dominated stands, *P. pseudostrobus* dominated forests account for 32.8% (18,462 ha) of the reserve, the co-dominated forests occupy 27.8% (15,078 ha) of the reserve, whereas only 12.7% of the surface (7153 ha) is dominated by *A. religiosa*. The rest of the reserve is covered by other vegetation types.

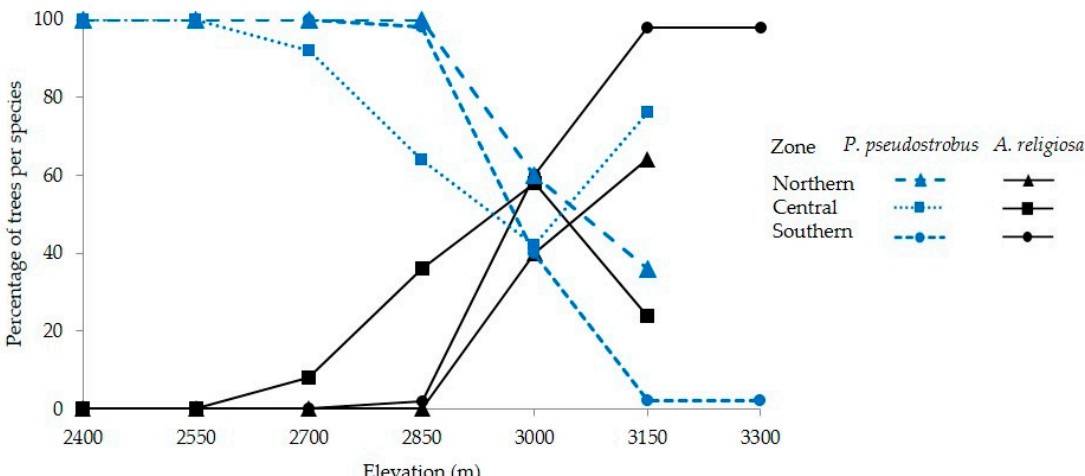

**Figure 3.** Proportion of *P. pseudostrobus* (broken lines) and *A. religiosa* (continuous lines) trees along an altitudinal gradient in the MBBR, in the northern (triangles), central (squares), and southern (circles) zones.

### 3.2. Establishment History of P. pseudostrobus and A. religiosa

The recorded ages ranged between 12 and 106 years for *A. religiosa* and between 10 and 120 years for *P. pseudostrobus*, although there were only three individuals over 100 years old in the latter species. Trees of *A. religiosa* were significantly older (average = 51 years old) in comparison to *P. pseudostrobus* trees (average = 39 years old) (df = 1, W = 101,730, $p < 0.05$).

Differences in the age of *A. religiosa* associated within the zones were found (df = 2, F = 19.86, $p < 0.05$), with younger trees in the central zone compared with those of the northern and southern zones. Age of *P. pseudostrobus* was affected by elevation (df = 1, F = 13.66, $p < 0.05$), zone (df = 2, F = 61.22, $p < 0.05$), and slope aspect (df = 1, F = 9.13, $p < 0.01$). For this latter species, age increased with altitude, while tree ages in the central zone were younger than in the southern and northern zones. In addition, trees in northern aspects were older than trees in southern-facing slopes.

The establishment of *A. religiosa* trees was continuous in dominated and co-dominated forests. The mono-dominant stands (altitude > 3300 m, southern area) showed continuous tree recruitment between 1920 and 1985, in which no more than one five-year class without establishment was recorded, and no recruitment pulses were observed (Figure 4a). In co-dominated stands (2850–3150 m), we observed continuous recruitment of *A. religiosa* trees, established from 1950 to 1995 (Figure 4b). At the stand at 3150 m in the southern zone, there was a period with high establishment between 1955 and 1975. However, that period was preceded and followed by continuous recruitment and was not associated with wet conditions. Thus, we determined that a continuous establishment with no pulses took place. Moreover, tree establishment was not correlated with wet years in both dominances.

The establishment of *P. pseudostrobus* was also continuous in co-dominated stands (Figure 4c). In the stand located at 3150 m, central zone, South-facing slope, and those located at 2850 m in the northern and central zones, there was at least one five-year period with apparent peaks in tree establishment. However, each of these two periods was preceded and succeeded by periods with tree recruitment. Hence, these visually-apparent pulses of recruitment did not meet the criteria for establishment pulses defined previously, and they were not associated clearly with wet years. Moreover, considering the establishment of both *A. religiosa* and *P. pseudostrobus* in co-dominant stands, we observed that these species established simultaneously, in contrast with the expected pattern in which *P. pseudostrobus* establish before *A. religiosa*.

In *P. pseudostrobus* dominated stands (Figure 4d), despite continuous regeneration, there were three sites with apparent recruitment peaks, characterized by young trees, and having at least a five-year period with a maximum establishment. These were the stand at 2700 m in the southern

zone, South-facing slope, and the stand at 2400 m in the northern zone, South-facing slope. However, before and after said peaks of establishment there were periods of tree recruitment, so the establishment pattern was considered continuous and visually not correlated with wet years.

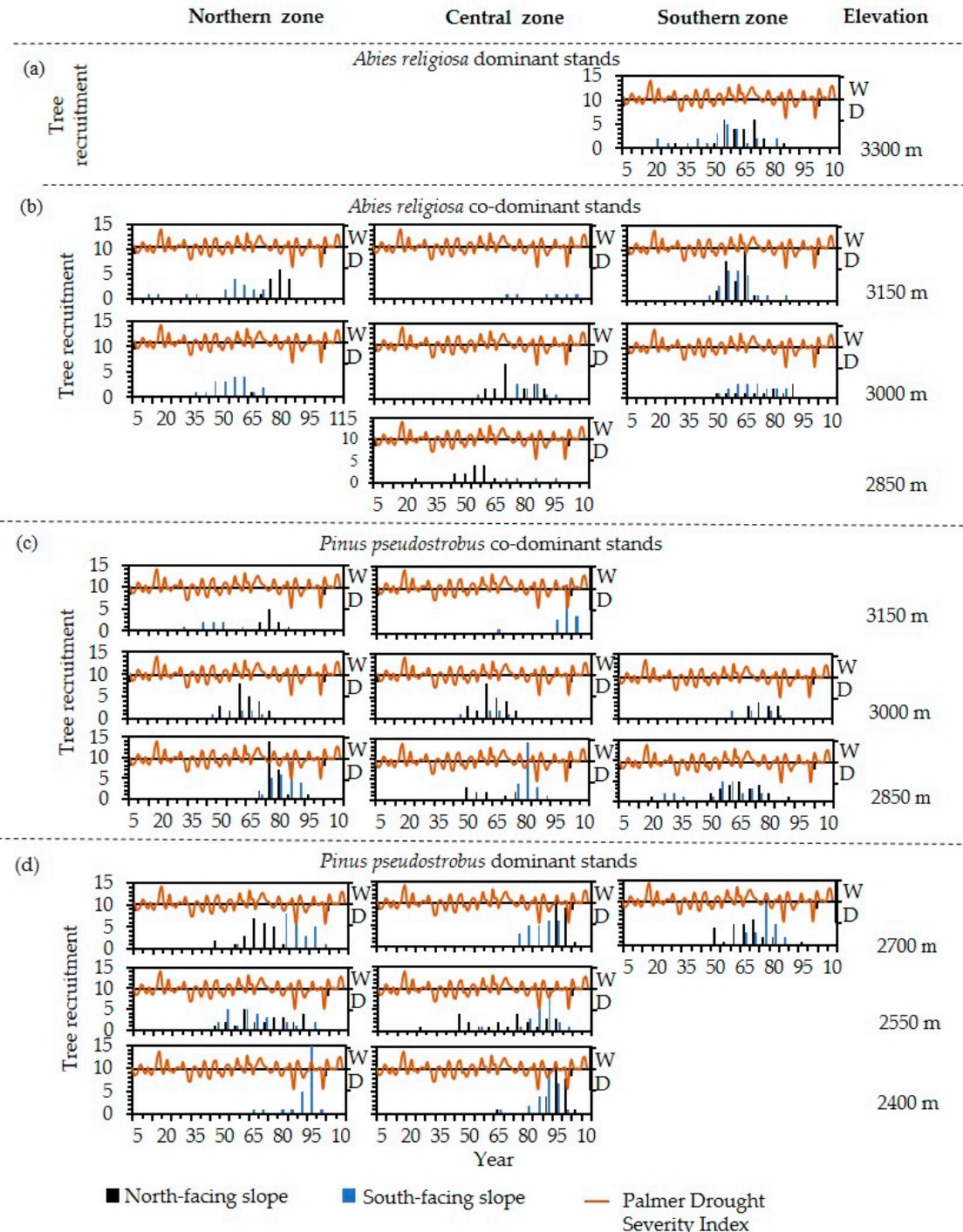

**Figure 4.** Establishment history in the MBBR, central Mexico, at different altitudes in (**a**) stands dominated and (**b**) co-dominated by *A. religiosa*, and in stands (**c**) co-dominated and (**d**) dominated by *P. pseudostrobus*. Years in which trees were established are plotted in 5-year intervals. Palmer Drought Severity Index values above line (W) indicate wet conditions, whereas values under the line (D) indicate dry conditions.

## 4. Discussion

Stands dominated and co-dominated by both species are distributed according to an altitudinal gradient in the MBBR. This elevation gradient is consistent with a climatic gradient where higher elevations are associated with higher humidity and lower temperatures, which facilitate the establishment of *A. religiosa*, while drier conditions at lower elevations promote the establishment of *P. pseudostrobus* [60]. This pattern is similar to the gradients of temperature, precipitation, and forest dominance found in montane conifer forests of the Sierra Nevada, California, and the Sky Islands, Arizona, in the US Southwest [9,61].

Although very few studies have assessed the altitudinal distribution of both species, this pattern is consistent with the elevation distribution of other Mexican *Pinus* and *Abies* species. Particularly, *Abies vejari* Martínez is a dominant species above 3000 m in the Sierra Madre Oriental in northeastern Mexico [62], *Abies hickelii* Flous and Gaussen is dominant between 3000 and 3600 m in the eastern Trans-Mexican Volcanic System [63], and co-dominated forests of *Pinus* sp. and *Abies* spp. were found between 2900 and 3300 m in northern Mexico [64]. Moreover, forests dominated by *P. pseudostrobus* were documented beneath 2800 m in central Mexico [52].

The continuous tree establishment in *A. religiosa* and *P. pseudostrobus* dominant and codominant stands confirms a constant demographic replacement, which in turn suggests that sacred fir forests, mixed-conifer forests, and smooth-bark Mexican pine forests experience similar disturbances regimes. Since this kind of forest is shaped and strongly influenced by fire [22], our data suggest that the conifer forests in the MBBR experience frequent, low-severity, and low-intensity fires, which do not trigger stand-replacement fires. These results are consistent with the fire regimes found for *P. pseudostrobus* and *A. religiosa* mono-dominated stands in the MBBR [23].

Most fire history reconstructions conducted in Mexican subtropical and tropical conifer forests have documented frequent, low-severity, and low-intensity fire regimes; for example, in Jeffrey pine forests in Sierra San Pedro Mártir, Baja California [65], in mixed-conifer forests in northwestern Mexico [31], in montane conifer forests in northeastern Mexico [66], as well as in high-elevation *Pinus hartwegii* Lindley forests in the Trans-Mexican Volcanic System [67]. Currently, it is unknown whether this fire regime is a unique feature of montane conifer forest in the Mexican subtropics and tropics, but it is possible that both the rapid replacement of fuel loads and the decay of forest legacies do not allow for their accumulation, and inhibit the generation of high-severity and high-intensity fires [68]. In consequence, the continuous gap formation by the occurrence of small fires could explain the continuous tree establishment observed in this study, even under dry years.

An unexpected finding was the young tree ages of both species since the majority of sacred fir and smooth-bark Mexican pine trees in the MBBR are no older than 120 years old. Although in Mexico the existence of stands dominated by millennial trees has been reported [69], many dendrochronological studies have found similar maximum ages of less than 150 years in Mexican tropical conifer species, such as *Pinus teocote* Schiede. Ex Schltdl and Cham [70], *Pinus oocarpa* Schiede ex. Schltdl [71] and *A. religiosa* [72]. Indeed, other ecosystem processes evaluated in Mexican tropical conifer forests, such as fast growth rates [73], the rapid decay of fuel loads [74], fast decomposition of forestry legacies [13], and frequent fire regimes [63] differ from those documented at higher latitudes [75,76]. In tropical regions, there is less seasonal variation in temperature and precipitation in comparison with boreal or austral conifer forests [77,78]. These factors could suggest that montane tropical conifer forests, such as those located in tropical Mexico, have a higher turn-over rate, hence explaining partially the presence of younger trees compared to higher-latitude conifer forests.

A second alternative to explain the continuous tree establishment and the relatively young tree ages of *P. pseudostrobus* and *A. religiosa* is related to the effect of human activities and the disruption of natural fire regimes. In the conifer forests of the MBBR, selective and illegal logging may have reduced the proportion of large trees [45], which in turn could have decreased the number of the oldest trees. The current firefighting and fire suppression [48] may have altered fuel loads dynamics and limited the spread, intensity, severity, and occurrence of large fires, especially in sacred fir forests,

which in turn could progressively increase fuel loads. However, fuel loads may have been reduced by persistent commercial logging, constant firewood collection, and livestock grazing, as well as by eventual salvage logging after windthrows, bark beetle detections, and fires, which may have also been opening gaps with sufficient light conditions to enhance a continuous tree establishment [79]. Hence, human disturbances may have replaced the role of fire in the regeneration of *A. religiosa* and *P. pseudostrobus* since potential stand-replacement fires have not influenced peaks of tree establishment.

The disruption of natural fire regimes in the MBBR could lead to negative consequences on conifer forests. Since our results suggest an increase in fire frequency in sacred fir forests, frequent fires can limit tree recruitment and raise tree mortality rates in species adapted to infrequent fires, such as *A. religiosa* [80,81]. In mixed-conifer forests, the increase of fire frequency may generate the replacement of fire-intolerant species, such as *A. religiosa*, for more fire-resistant species, such as *P. pseudostrobus* [82]. Both scenarios imply the reduction of the trees where the monarch butterflies preferentially perch, which represents a serious risk for this migratory species. Concerning smooth-bark Mexican pine forests, the current policy of fire suppression could enhance the progressive accumulation of fuel loads and subsequent high-severity and high-intensity fires, as documented in many dry montane forests of the US Southwest [79].

Another unexpected finding was the similar ages found between *A. religiosa* and *P. pseudostrobus* trees in mixed stands. It is expected that, as a consequence of global warming, *A. religiosa* populations will eventually migrate to upper elevations, while populations of *P. pseudostrobus* will replace them [34]. If this process were happening in the MBBR, we should have documented young smooth-bark Mexican pine and old sacred fir trees in co-dominated stands. However, our data do not confirm this displacement as the climate warms. This result suggests that the MBBR could be acting as a climatic relict, in which temperature shifts have not yet been intense enough to lead an upper migration [83]. In further research, it would be important to complement the information of this study with trees ≤25 cm DBH, so the younger recruits in the populations of both species can be recorded.

Furthermore, the ages of *P. pseudostrobus* and *A. religiosa* overlapped, suggesting that both species have simultaneously been establishing across the reserve, which contradicts the route of succession proposed for Mexican tropical mixed-conifer forests [36]. This pattern may be explained by the short cone production cycles (2–3 years) in both species, typical of many conifers [52,84], and low-extent gaps produced by fires, which together can allow a rapid and simultaneous establishment. A better understanding of tree establishment after fires in mixed conifer stands is needed, but this is the first indication of a unique successional pattern in tropical mixed-conifer forests.

Hence, more research on disturbance regimes and their effect on tree populations is needed in the MBBR, as well as in Mexican tropical montane conifer forests. This study demonstrates that the reconstruction of establishment history, based on the age structure of tree populations, provides useful insight into the potential modification of disturbance regimes in conifer forests. The information derived from this kind of analysis is fundamental to design forest management recommendations consistent with natural disturbance regimes in these tropical montane forests.

## 5. Conclusions

Tree populations of *A. religiosa* and *P. pseudostrobus* are distributed following an altitudinal gradient, in which tree stands dominated by *A. religiosa* are located at elevations higher than 3150 m, codominant stands between 2850 and 3150 m, and stands dominated by *P. pseudostrobus* at elevations lower than 2850 m.

The age structure of both species showed a continuous establishment pattern that suggests that forest dominated or co-dominated by *A. religiosa* and *P. pseudostrobus* experience similar fire regimes, characterized by frequent, low-severity, and low-intensity fires. This tree establishment pattern, coupled with young tree ages, suggests, on the one hand, that natural fire regimes have been altered by the effect of human disturbances, influencing age structure, fuel load dynamics, but not inhibited the

regenerative capability of tree populations. On the other hand, these fire regimes and demographic dynamics are likely a singular feature of these tropical montane conifer forests.

It is necessary to assess the effect of natural and human disturbances on age structure and fire regimes in conifer forests in the MBBR. This study was a first step in analyzing fire regimes through age structure analysis. This information is useful to suggest management of natural and human disturbances, to preserve the integrity of the overwintering habitat of the monarch butterflies.

**Author Contributions:** Conceptualization, methodology, formal analysis, investigation, writing—original draft preparation, writing—review, and editing were conducted by both J.E.S.-C. and D.R.P.-S. All authors have read and agreed to the published version of the manuscript.

**Funding:** This research was funded by SEP-CONACYT, 2010-154434, project "Effect of natural and human disturbances in conifer forests of the Monarch Butterfly Biosphere Reserve: implications for fire management".

**Acknowledgments:** The first author thanks the Graduate Program in Biological Sciences of the Universidad Nacional Autónoma de México (UNAM) for the support to conduct his graduate studies, and the National Council of Science and Technology (CONACYT) for the scholarship received during his studies. The authors thank the Institute for Ecosystem Research and Sustainability (IIES-UNAM), the Secretariat of Environment and Natural Resources (SEMARNAT), the authorities of the MBBR (CONANP) and communal authorities of ejidos Chincua, Jesús Nazareno, Hervidero y Plancha, Las Trojes, El Asoleadero, San Juan Zitácuaro, Carpinteros, and Vare Chiquichuca, for the permission and support to conduct fieldwork. Special thanks to Miguel Martínez, Teresa Valverde, and J. Trinidad Sáenz for their suggestions for the development of the study. Anastasio Sarmiento and Gerardo Segundo, Álvaro González, Diego González, Javier Colin, and Luis Lara provided technical support during field sampling, and the Eco-garden at IIES-UNAM facilitated space to sand the sample cores. Finally, we thank David Ruiz, Gabriela Trejo, and Mary-Ann Hall for the editorial revision of the manuscript.

**Conflicts of Interest:** The authors declare no conflict of interest.

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
