# Peer review of "Modification of Fire Regimes Inferred from the Age Structure of Two Conifer Species in a Tropical Montane Forest, Mexico"

_forests, doi:10.3390/f11111193_

Round 1
Reviewer 1 Report
This was an excellent study and well-written. Dealing with fire regime reconstructions in such a setting is problematic given extensive logging and landscape management in the region, but the authors do a thorough job of explaining those influences and how they may impact their study design. The discussion of the data was thoughtful and not "over-interpreted". The dataset is also new and important from such an understudied region, so I think this will have broad impact, especially if the data is made available in an online archive following publication.
My only major comment is this: What are the differences in fire-adapted traits between these two species? For example, some tree species support crown fires and allocate little biomass to their lower branches. Other trees support lower severity ground fires by pruning their lower branches. Still others have have serotinous cones that only open with fire, or resinous leaves that are more flammable, etc. If there is any information about such fire adapted traits (and/or if they differ between the two taxa), then it should be discussed here. By investing a bit of text into discussing the different types of fire regimes expected to be supported with these two tree species, it could potentially bolster the interpretation of the data.
Author Response
Response: Thank you for your comments and your suggestion. We added information about fire-adapted traits in both species. Particularly, sacred fir trees form dense canopies, and their branches frequently reach the ground, which suggests that sacred firs are not subjected to frequent fires (Rodríguez-Trejo, 2008). In contrast, smooth-bark Mexican pines have thick bark and high crown bases, traits associated with frequent fires (Rodríguez-Trejo and Fulé, 2003).
Reviewer 2 Report
Thank you for the opportunity to review your manuscript. This work is very valuable due to its location in a globally important area with limited previous study. Please forgive any oversights on my part—you may have already addressed some of these points sufficiently and should feel free to point that out. As an aside, only if you have the time and the journal editors are agreeable, you might consider including a Spanish translation of the paper as a supplement.
Although I have a lot of comments, these are suggestions and should be incorporated at the authors' discretion. The only substantial change, if accepted, would be incorporating scPDSI in the graphical presentation of results in Figure 4.
I have organized my comments by section, below.
- introduction—Here are my suggestions for reorganization and a few pieces of additional information.
- Start with the paragraph “Conifer forests distributed…” (line 65) and move the next two paragraphs up with this one. I think starting with this information will really draw the reader in.
- Then set the stage for your questions by giving background information from your paper in Fire Ecology (Sáenz-Ceja, J.E., Pérez-Salicrup, D.R. Dendrochronological reconstruction….) including what you learned and what others have learned about the MBBR forests. Your previous work made conclusions about the influence of people overriding climate and driving fire regime traits that should be mentioned here. And I think you are also looking for confirmation or contradiction to what was said about tree regeneration not being associated with cyclical fire events (in “The role of fire…” and in Perez-Salicrup et al. 2016). Basically, in this new paper you are following up on what you learned by using a somewhat different methodology to retest your hypotheses regarding episodic versus continuous regeneration across the elevational gradient that would be expected due to fire.
- Introduce the idea that fires influence age structure and thus quantifying age structure may give additional information about fire regimes along the lines of what you’ve written, but I’d recommend that you broaden the geographic scope of your citations that back up this idea—use one or two from the US southwest but also insert a few that represent broader geography (see literature suggestions, below—and you can probably find others). The coniferous species you are studying may be similar in some ways to trees in the US southwest, but the species traits (that relate to fire regimes/adaptations) and the environment are unique.
- Introduce the tree species, as you have done, but it would also be helpful to know what their life history might tell you about their fire history/adaptations. For the true firs, species traits may be fairly generalizable across regions—but you can describe those specifically for A. religiosa. For Pinus, a great diversity of traits can be present—and pseudostrobis is one that is less well known outside of your region of study. Can this Pinus species be classified in terms of the syndromes described by Keeley 2012 (see below) including, unique life stages (e.g., grass-stage seedlings)? This background is important for discussion of your assumptions about fire regimes in these forests and factors contributing to successful regeneration e.g., dispersal mode.
- Methods
- Very complete and easy to follow.
- I’d suggest you add the PDSI trend line to the graphs in Figure 4. I noticed you didn’t observe much variability in climate indices in your earlier analysis of tree rings, but you might do better with the spatial data described here: https://agupubs.onlinelibrary.wiley.com/doi/10.1002/jgrd.50355 and available here: https://crudata.uea.ac.uk/cru/data/drought/. The data can be sampled for your location and years/months of interest and may better illustrate patterns specific to your region and timeframe.
- Results
- Figure 3. The legend is mis-labeled—the species names are switched!
- Figure 4. I’m sure this will be much bigger in the final version—right now it’s difficult to read but I assume that is due to the need to paste it within the text.
- If you follow up on the suggestion, above and look at scPDSI, do the trends in scPDSI tell you something interesting about climate conditions during the periods of regeneration that might relate to fire and/or other processes conducive to regeneration?
- Discussion
- Your observations on shifts in distribution expected with climate warming are very important and somewhat reassuring. Do you think this may reflect the status of these forests as climate relicts? See Hampe and Jump 2011.
- Can you cite where your results confirm or how they shed new light on fire regimes compared to your study in Fire Ecology? I think you already have made these points in the discussion, just need to cite your paper.
- Your discussion of various factors influencing fire regimes and tree regeneration is very compelling. Regarding lack of stand replacement fires, it seems likely that fuel continuity between lower elevation pines and upper elevation fir forests is a factor. If lower elevation forests were intact, frequent fires there would spread into higher elevations when dry conditions occurred. The clearing of pine forests along with the setting of fires by people have fragmented the fuels so that even in episodically dry years, fire does not spread to upper elevations where high severity could be more likely. This goes along with your observation of continuous gap formation. I appreciate all the points you have made along these lines.
Do any of these ideas spark questions for future research (that you haven’t already mentioned!)? Here are some topics you might incorporate or embellish upon—use as you see fit!
- microenvironmental affinities for germination and growth and how these interact with fire https://www.frontiersin.org/articles/10.3389/fevo.2019.00421/full (restoration, assisted migration)
- does consideration of species traits (dispersal ability, adaptations to survive fire…) alter your previous assumptions about fire history, or do the traits provide further evidence for those assumptions?
- how might expectations differ in mixed stands, given life history traits (mix of pines that may be fire resisters and firs that are not survivors but are more geared toward post-fire dispersal may result in mixed fire regime rather than frequent fire regime?)
- Future research needs: tracking contemporary fire history (field records and remotely sensed data). Following up on questions raised in Cuauhtémoc Sáenz-Romero et al.
Additional literature—may not all be relevant but wanted to share in case you find useful.
Mentioned above
Carbajal-Navarro, A., Navarro-Miranda, E., Blanco-García, A., Cruzado-Vargas, A. L., Gómez-Pineda, E., Zamora-Sánchez, C., … Sáenz-Romero, C. (2019). Ecological Restoration of Abies religiosa Forests Using Nurse Plants and Assisted Migration in the Monarch Butterfly Biosphere Reserve, Mexico. Frontiers in Ecology and Evolution, 7, 421. https://doi.org/10.3389/fevo.2019.00421
Hampe, A., & Jump, A. S. (2011). Climate relicts: Past, present, and future. Annual Review of Ecology, Evolution and Systematics, 42, 313–333. https://doi.org/10.1146/annurev-ecolsys-102710-145015
Keeley, J. E. (2012). Ecology and evolution of pine life histories. Annals of Forest Science, 69(4), 445–453. https://doi.org/10.1007/s13595-012-0201-8
Other factors: seed availability
Andrus, R. A., Harvey, B. J., Hoffman, A., and Veblen, T. T.. 2020. Reproductive maturity and cone abundance vary with tree size and stand basal area for two widely distributed conifers. Ecosphere 11( 5):e03092. 10.1002/ecs2.3092
References from various regions of study
Pyke, C.R., Condit, R., Aguilar, S. and Lao, S. (2001), Floristic composition across a climatic gradient in a neotropical lowland forest. Journal of Vegetation Science, 12: 553-566. doi:10.2307/3237007
Regeneration in mixed pine-oak forests: https://www.x-mol.com/paper/1254921113097363456
Disturbance and climate in mixed forests: https://link.springer.com/article/10.1007/s10021-019-00462-x
Grau, H.R. and Veblen, T.T. (2000), Rainfall variability, fire and vegetation dynamics in neotropical montane ecosystems in north‐western Argentina. Journal of Biogeography, 27: 1107-1121. doi:10.1046/j.1365-2699.2000.00488.x
BUNKER, D.E. and CARSON, W.P. (2005), Drought stress and tropical forest woody seedlings: effect on community structure and composition. Journal of Ecology, 93: 794-806. doi:10.1111/j.1365-2745.2005.01019.x
Hansen, W. D., Abendroth, D., Rammer, W., Seidl, R., and Turner, M. G.. 2020. Can wildland fire management alter 21st‐century subalpine fire and forests in Grand Teton National Park, Wyoming, USA? Ecological Applications 30( 2):e02030. 10.1002/eap.2030
Author Response
Comments and Suggestions for Authors
Thank you for the opportunity to review your manuscript. This work is very valuable due to its location in a globally important area with limited previous study. Please forgive any oversights on my part—you may have already addressed some of these points sufficiently and should feel free to point that out. As an aside, only if you have the time and the journal editors are agreeable, you might consider including a Spanish translation of the paper as a supplement.
Although I have a lot of comments, these are suggestions and should be incorporated at the authors' discretion. The only substantial change, if accepted, would be incorporating scPDSI in the graphical presentation of results in Figure 4.
I have organized my comments by section, below.
- introduction—Here are my suggestions for reorganization and a few pieces of additional information.
- Start with the paragraph “Conifer forests distributed…” (line 65) and move the next two paragraphs up with this one. I think starting with this information will really draw the reader in.
- Response: We decided to maintain the current paragraph structure because we tried to start from the general framework of age structure and fire regimes in conifer forests and then explain conditions at the study site. However, we moved the paragraph that mentioned the consequences of fire regime modifications, which now follows the paragraph in which we mention the importance of conducting this research in the Monarch Butterfly Reserve.
- Then set the stage for your questions by giving background information from your paper in Fire Ecology (Sáenz-Ceja, J.E., Pérez-Salicrup, D.R. Dendrochronological reconstruction….) including what you learned and what others have learned about the MBBR forests. Your previous work made conclusions about the influence of people overriding climate and driving fire regime traits that should be mentioned here. And I think you are also looking for confirmation or contradiction to what was said about tree regeneration not being associated with cyclical fire events (in “The role of fire…” and in Perez-Salicrup et al. 2016). Basically, in this new paper you are following up on what you learned by using a somewhat different methodology to retest your hypotheses regarding episodic versus continuous regeneration across the elevational gradient that would be expected due to fire.
Response: We added information from our previous article “Dendrochronological reconstruction…”, particularly on how human disturbances could have influenced the current fire regime in the MBBR.
- Introduce the idea that fires influence age structure and thus quantifying age structure may give additional information about fire regimes along the lines of what you’ve written, but I’d recommend that you broaden the geographic scope of your citations that back up this idea—use one or two from the US southwest but also insert a few that represent broader geography (see literature suggestions, below—and you can probably find others). The coniferous species you are studying may be similar in some ways to trees in the US southwest, but the species traits (that relate to fire regimes/adaptations) and the environment are unique.
Response: We agree that montane tropical forests in Mexico share ecological characteristics with subtropical montane forests, such as those located in the US Southwest. Therefore, we added references to studies conducted in the US Southwest (Fulé et al., 2003; Fulé y Laughlin, 2007).
- Introduce the tree species, as you have done, but it would also be helpful to know what their life history might tell you about their fire history/adaptations. For the true firs, species traits may be fairly generalizable across regions—but you can describe those specifically for A. religiosa. For Pinus, a great diversity of traits can be present—and pseudostrobus is one that is less well known outside of your region of study. Can this Pinus species be classified in terms of the syndromes described by Keeley 2012 (see below) including, unique life stages (e.g., grass-stage seedlings)? This background is important for discussion of your assumptions about fire regimes in these forests and factors contributing to successful regeneration e.g., dispersal mode.
Response: As suggested by you and the first reviewer, we added information about fire-adapted traits of sacred firs and smooth-bark Mexican pines. Sacred fir trees exhibit a low crown base and form dense stands, which can be associated with infrequent fires, whereas the pines of the MBBR have thick bark and low crown bases, typical traits of trees adapted to frequent fires.
- Methods
- Very complete and easy to follow.
- I’d suggest you add the PDSI trend line to the graphs in Figure 4. I noticed you didn’t observe much variability in climate indices in your earlier analysis of tree rings, but you might do better with the spatial data described here: https://agupubs.onlinelibrary.wiley.com/doi/10.1002/jgrd.50355 and available here: https://crudata.uea.ac.uk/cru/data/drought/. The data can be sampled for your location and years/months of interest and may better illustrate patterns specific to your region and timeframe.
- Response: We followed your suggestion and added the PDSI values in Figure 4. These values were reconstructed by Stahle et al 2016, in a near-site located 100 km from the MBBR. Therefore, it is likely that this index represents past climatic conditions that covered the MBBR extent.
- Results
- Figure 3. The legend is mis-labeled—the species names are switched!
- Response: Thank you for pointing this mistake. We corrected it.
- Figure 4. I’m sure this will be much bigger in the final version—right now it’s difficult to read but I assume that is due to the need to paste it within the text.
- If you follow up on the suggestion, above and look at scPDSI, do the trends in scPDSI tell you something interesting about climate conditions during the periods of regeneration that might relate to fire and/or other processes conducive to regeneration?
Response: Previously, we had compared the establishment years with a precipitation index reconstructed from tree rings of sacred firs and smooth-bark Mexican pines in the MBBR (Carlón-Allende et al., 2016, “Climatic responses of Abies religiosa and Pinus pseudostrobus in the MBBR”). However, no trends were identified. In this version, using PDSI values, we do not observe a visual correlation between tree establishment and wet periods. Furthermore, tree recruitment also took place during dry years. As we mentioned in our previous article, “Dendrochronological reconstruction…”, climatic conditions apparently do not strongly influence fire return intervals nor tree recruitment. This study confirms that statement.
- Discussion
- Your observations on shifts in distribution expected with climate warming are very important and somewhat reassuring. Do you think this may reflect the status of these forests as climate relicts? See Hampe and Jump 2011.
Response: After reading the article of Hampe and Jump 2011, we think that it is likely that the MBBR may be acting as a climate relict. As mentioned above, dry years do not trigger fires nor wet periods enhanced tree establishment. It is possible that, due to the low-contrasting temperature and precipitation regimes in tropical habitats, climate variability does not influence strongly ecological processes, such as fires, tree recruitment, and upper migration, as documented in higher latitudes.
- Can you cite where your results confirm or how they shed new light on fire regimes compared to your study in Fire Ecology? I think you already have made these points in the discussion, just need to cite your paper.
Response: Thank you for the suggestion. We cited our article in Fire Ecology, as a confirmation of our results obtained in this study.
- Your discussion of various factors influencing fire regimes and tree regeneration is very compelling. Regarding lack of stand replacement fires, it seems likely that fuel continuity between lower elevation pines and upper elevation fir forests is a factor. If lower elevation forests were intact, frequent fires there would spread into higher elevations when dry conditions occurred. The clearing of pine forests along with the setting of fires by people have fragmented the fuels so that even in episodically dry years, fire does not spread to upper elevations where high severity could be more likely. This goes along with your observation of continuous gap formation. I appreciate all the points you have made along these lines.
Response: Thank you for your comment. We are trying to show the ecology community that ecological processes in tropical montane forests may be substantially different from those located in higher latitudes by both a biogeographical aspect and the influence of human disturbances.
- Do any of these ideas spark questions for future research (that you haven’t already mentioned!)? Here are some topics you might incorporate or embellish upon—use as you see fit!
Response: Thank you so much for these suggestions. Information on these topics is scant in the forests of the MBBR.
- microenvironmental affinities for germination and growth and how these interact with fire https://www.frontiersin.org/articles/10.3389/fevo.2019.00421/full (restoration, assisted migration)
- does consideration of species traits (dispersal ability, adaptations to survive fire…) alter your previous assumptions about fire history, or do the traits provide further evidence for those assumptions?
- how might expectations differ in mixed stands, given life history traits (mix of pines that may be fire resisters and firs that are not survivors but are more geared toward post-fire dispersal may result in mixed fire regime rather than frequent fire regime?)
- Future research needs: tracking contemporary fire history (field records and remotely sensed data). Following up on questions raised in Cuauhtémoc Sáenz-Romero et al.